# TOWARDS UNDERSTANDING DATA VALUES: EMPIRICAL RESULTS ON SYNTHETIC DATA

## ABSTRACT

Understanding the influence of data on machine learning models is an emerging research field. Inspired by recent work in data valuation, we perform several experiments to get an intuition for this influence on a multi-layer perceptron. We generate a synthetic two-dimensional data set to visualize how different valuation methods value data points on a mesh grid spanning the relevant feature space. In this setting, individual data values can be derived directly from the impact of the respective data points on the decision boundary. Our results show that the most important data points are the miss-classified ones. Furthermore, despite performance differences on real world data sets, all investigated methods except one qualitatively agree on the data values derived from our experiments. Finally, we place our results into the recent literature and discuss data values and their relationship to other methods.

## 1 INTRODUCTION

Machine learning algorithms stand and fall with the training data. Although the process of data collection and labeling is highly time consuming, the creation of quality training data sets is of paramount importance. However, it turns out that not all data points contribute equally to the quality of the trained models. Recently, several works have addressed the problem of measuring the value of individual data points for the performance of the algorithm such as data Shapley (Ghorbani & Zou, 2019), catastrophic forgetting (Toneva et al., 2018), influence functions (Koh & Liang, 2017), and data valuation using reinforcement learning (DVRL, (Yoon et al., 2019)).

In this paper we address the question: what distinguishes a high value data point from a low value one according to the state of the art works listed above? Accordingly, our goals are to support a better understanding of the models and a more efficient data collection.

Our contributions can be summarized as follows: We propose a framework to describe data values as the resulting change of a model's decision boundary with respect to added data points. Subsequently, we probe the above data evaluation methods using the introduced framework on a synthetic 2D data set and visualize the resulting data values. Finally, we discuss the implications of our findings and place them in the recent literature.

### 1.1 BACKGROUND

There are several directions of research that can be interpreted as being related to estimating the value of data. *Active Learning* (Settles, 2010) and *core-set* estimation (Mirzasoleiman et al., 2020; Sener & Savarese, 2018) historically belong to the earlier approaches. Points chosen for labeling in active learning and core-sets should intuitively have high data values. On top of that, recent methods such as influence functions (Koh & Liang, 2017), DVRL (Yoon et al., 2019) and data Shapley (Ghorbani & Zou, 2019) directly aim at estimating the value of data. Finally, several other works are related or can be applied to the estimation of data values. These works include forgetting events discovered by Toneva et al. (2018) and memorization effects described by Feldman (2020b) and Jiang et al. (2020). Another related work by Paul & Dziugaite (2021) deals with estimating important samples early in training. In this paper, we empirically compare four of these methods on a synthetic data set in order to develop an intuition of how they work and to understand what makes a point important.

To measure the importance of data points we consider two definitions. An important point either causes a large absolute change in the model's estimated decision boundary, or reduces the relative distance between the estimated decision boundary and the ground-truth decision boundary. The reason to consider both is that some points could result in a large change of the decision boundary while increasing the error on the test set (e.g., a miss-labelled point). However, access to the ground-truth decision boundary is rarely available in practice. As our results on noise-free data show, there is no big difference between these two definitions, provided that the train data set is free of label noise.

## 1.2 RELATED WORK

In this section, we briefly outline a few research fields related to data valuation and revisit the four different data valuation methods from our experiments more closely. We use the standard notation to refer to the data set $D = \{(x_i, y_i)\}_{i=0}^n$ and sometimes the shorter form $D = \{X, y\}$.

**Core-Set Estimation**   Historically, core-set estimation is one of the first approaches that indirectly address data values (Feldman, 2020a). The goal in core-set estimation is the selection of a small core-set $D^C \in D$ resulting in the same or similar performance of the model as if it was trained on the entire training set $D$. Many core-set-selection methods are model dependent. A recent work by Sener & Savarese (2018) applies core-sets to active learning and convolutional neural networks. As mentioned before, the connection between core-sets and data values is that one would expect points in the core-set to be of high data value.

**Active Learning**   Another related field is active learning (Settles, 2010). Given a labeled data set $\{X, y\}$, and a large set of unlabeled points $X^U$, the algorithm can query an oracle for the label of a number of unlabeled data points $x_i^U \in X^U$ up to some budget $b$. An easy baseline for active learning is to select data points predicted with low confidence score. We, therefore, evaluate some out-of-distribution methods (Lakshminarayanan et al., 2017; Gal & Ghahramani, 2016) on the task of data valuation in Appendix A.2. A link between active learning and data valuation can be established quite naturally: the points chosen by active learning methods should also have a high data value.

**Leave-One-Out Valuation (LOO)**   A straight forward way of estimating the impact of data points is cancelling individual data points iteratively and comparing model performances after training. Let $f_X$ refer to the model trained on the entire train set $X$ and $f_{X \setminus x_i}$ on the train set without $x_i$. The data value of $x_i$ is then directly given by the performance difference of $f_X$ and $f_{X \setminus x_i}$ on the test set.

**Influence Function**   Influence Functions were proposed by Koh & Liang (2017) as a way of measuring the influence a data point will have on the model without retraining. As Koh & Liang (2017) propose, influence functions should be an approximation for leave-one-out retraining. The influence is estimated by the change in the model parameters $\theta$. That is, the influence of data point $z_i = (x_i, y_i)$ is given by $\hat{\theta}_{-z_i} - \hat{\theta}$, where $\hat{\theta}_{-z_i}$ refers to the model parameters if the point $z_i$ was not part of the training data.

Since our synthetic data set is small, we can afford retraining and do not consider influence functions.

**Data Shapley**   Ghorbani & Zou (2019) criticize that leave-one-out valuation does not capture interdependence between points. They propose data Shapley as a way of tackling this. The value of a data point $x_i$ is measured by its average contribution to the model performance when trained on all subsets of the remaining points $X \setminus x_i$. Since this means exponential complexity, Ghorbani & Zou (2019) propose a truncated Monte Carlo scheme to estimate the value.

**Sample Forgetting**   Toneva et al. (2018) found that data points frequently forgotten during training are also important for training performance. A point is considered as forgotten if it was correctly classified at some time step $t$ during training but again miss-classified later. They also report that this scheme of finding important points works between different model architectures.

**Memorization**   is another related line of work. Feldman (2020b) and Jiang et al. (2020) independently use it for two different approaches. A training instance $x_i$ is referred to as singleton or rare

if removing $x_i$ from the train set reduces the probability for $x_i$ to be classified correctly. That is, $x_i$ is only memorized if it is part of the train data. Formally, $P\left(f_X(x_i) = y_i\right) \gg P\left(f_{X \setminus x_i}(x_i) = y_i\right)$. Feldman uses it to support the long-tail theory in Feldman (2020b) and provides empirical evidence for the theory in Feldman & Zhang (2020). In short, long tail refers to the fact that most data sets contain a long tail with many atypical instances. According to the theory, these examples are important for the generalization of the algorithm. Jiang et al. (2020) use memorization as a measure to characterize the regularities of a data set. They find that removing examples with the highest irregularity can improve training performance. These examples typically are the miss-labeled ones.

**DVRL**  Yoon et al. (2019) propose reinforcement learning as a way of estimating the value of data points. They train a data valuator to output values for each instance $(x_i, y_i)$ in the training set, and use the resulting data values to sample training batches. Each points' probability to be in the batch is proportional to this data value. A predictor model $f_B$ is trained on the selected batch $B$ and the resulting performance of the predictor is evaluated on a separate *target* set. The performance of the predictor model serves in turn as reward for the reinforcement learning of the data valuator. The valuator thereby learns to estimate important samples in the training set for the target distribution represented by the target set $\{X^V, y^V\}$. The data valuator (*DVRL*) is a function of $\mathrm{dvrl}\left(x_i, y_i, (f_X(x_i) - y_i)\right) \to \mathbb{R}$. In some initial experiments we were interested in whether we can remove the true labels $y_i$ from the DVRL while retaining the benefits to some extend but were not successful. Nevertheless, we provide results without labels in our experiments.

## 2  CONTRIBUTIONS

In this paper we use a synthetic 2D data set to visualize the importance of data points on the model performance. We span a mesh grid over the relevant feature plane and empirically compare the estimated data values of some of the methods introduced above. A comprehensible metric is then implemented to measure the difference between two decision boundaries. In the following section we first introduce the data set, describe how we estimate data values and, finally, explain our experimental setup.

### 2.1  DATA SET

The synthetic data set $D = \{(x_i, y_i)\}_{i=0}^n$ is two dimensional ($x_i \in \mathbb{R}^2$) and binary labeled ($y_i \in \{0, 1\}$). We randomly divided it into train and test split. It is sampled from a mixture of Gaussians for each label class and ground-truth labels are assigned by means of one-nearest-neighbor classification. We will refer to this ground-truth decision boundary as **g**. The data is not centered around the origin in order to have a non-linear decision boundary learned from the model and, thus, to make the graphs more comprehensible. On data centered in the origin a multi-layer perceptron learns linear decision boundaries as visible in the appendix in Figure 14. A detailed description of the data can be found in Appendix A.1. Furthermore, the data set $D^M = \{X^M, y^M\} = \{(x_i^M, y_i^M)\}_{i=0}^{m^2}$ is a 2D mesh grid used for plotting and estimating data values. For values $\{a_1, ..., a_m\}$ and $\{b_1, ..., b_m\}$ on the $x$ and $y$ axes, the mesh grid contains all pairs of values: $X^M = \{(a_1, b_1), (a_1, b_2), ..., (a_m, b_m)\}$ with corresponding labels $y^M \in \{0, 1\}^{m^2}$. Labels are computed with the ground-truth decision boundary **g**. In all experiments, the mesh grid is evenly spaced. The model will be referred to as $f_T$, where $T = \{(x_i^T, y_i^T)\}_{i=0}^{n_t}$ is the data set it was trained on. Hence, $f_T = f_T(x, \theta) = \arg\min_\theta \frac{1}{n_t} \sum_{i=0}^{n_t} L(f(x_i^T), y_i^T)$, where $L(.)$ is the loss function.

### 2.2  MEASURING DATA VALUES

For measuring data values we utilize both the ground-truth decision boundary **g** and a baseline model $f_X$ trained on the entire train data $X$. We can measure the influence of a point by the area between curves of the ground-truth decision boundary and the learned model. The mean absolute error (*MAE*) between two models $f_A$ and $f_B$ on the mesh grid approximates this area. Hence, we simply replace the test set with our mesh grid. The data value of a point $x_i$ is given by

$$\mathrm{dv}(f_{A(x_i)}, f_B, X^M) = \mathrm{MAE}\big(f_{A(x_i)}(X^M), f_B(X^M)\big) = \frac{1}{m^2} \sum_{k=0}^{m^2} \big|f_{A(x_i)}\big(x_k^M\big) - f_B\big(x_k^M\big)\big| \ .$$
(1)

Here, $f_{A(x_i)}$ is a model of interest (e.g. trained without $x_i$) and $f_B$ is a baseline. The sum runs over $m^2$ because it is the size of the mesh grid $\{(x_i^M, y_i^M)\}_{i=0}^{m^2}$. The leave-one-out data value for a single point $x_i$ would therefore be given by $\mathrm{dv}(f_{X \setminus x_i}, f_X, X^M)$.

**Advantage of the Decision Boundary Difference**   The advantage of using the change in the decision boundary to estimate data values is that model variations can be detected even if the accuracy stays constant. For instance, the change in the decision boundary could affect a region not represented by the original test data set. For 2D data using a mesh grid is still computable, but with increasing dimension it would become infeasible.

## 2.3 EXPERIMENTAL SETUP

This section first describes the model used in our experiments. Afterwards, the particular setup of the four algorithms we evaluate are presented. Furthermore, the appendix provides additional information regarding some baseline experiments and about the parameters used.

**Model**   In all experiments we use a multi-layer perceptron (MLP) with a single hidden layer of size 1,000. From our pre-experiments in Figure 7 of the appendix we selected this model because it had the highest stability and makes sure that data values are not disturbed by noise in the models' output. Furthermore, we use the MLP provided by *sklearn* (Buitinck et al., 2013) whenever possible. Only when necessary for the implementation, we use TensorFlow (Abadi et al., 2015).

**Leave-One-Out and Data Shapley**   We evaluate two different regimes using data Shapley and leave-one-out valuation. First, we evaluate the algorithms on the mesh grid $X^M$ only. Second, we evaluate the mesh grid w.r.t. $X$ as baseline. For the first, we estimate data values $\mathrm{dv}(f_{X^M \setminus x_i^M}, f_B, X^M)$ for each data point $x_i^M$ and w.r.t. some reference model $f_B$. For the latter we append points $x_i^M \in X^M$ sequentially to $X$ and evaluate $\mathrm{dv}(f_{X \cup x_i^M}, f_B, X^M)$. To compute the value with Shapley, there is an additional loop over the power set. For instance, the formula for the second regime is given by $\mathrm{dv}_{shap}(x_i^M, f_B) = \sum_{T \in \mathbb{P}(X^M)} \mathrm{dv}(f_{T \cup x_i^M}, f_B, X^M)$. In the appendix we further provide data values on the training data $X$ as $\mathrm{dv}(f_{X \setminus x_i}, f_B, X^M)$ .

**Sample Forgetting**   We use the same schemes as in leave-one-out and Shapley to estimate data values. In the first case we train $f_{X^M}$ and count how often each instance $x_i^M \in X^M$ was forgotten. In the second case we need to fit a model $f_{X \cup x_i^M}$ for each $x_i^M \in X^M$ once. The data value of $x_i^M$ is then given by the number of forgetting events of $x_i^M$ while fitting $f_{X \cup x_i^M}$. We scale the value to $[0, 1]$ for plotting. The procedure is described in Algorithm 1 of the appendix. Additionally, we provide results with *last-learned* samples, where we use the epoch a point was correctly classified in for the first time as data value. Again, results on the training data $X$ can be found in the appendix.

**Memorization**   For memorization we fit two models $k$-times and compute the difference between how often each $x_i^M$ was memorized correctly in each model. As before, we compare $f_{X \cup x_i^M}$ to $f_X$ in the main paper and provide $f_X$ to $f_{X^M \setminus x_i^M}$ in the appendix.

**DVRL**   We evaluate DVRL in three settings. With both ground-truth labels and predictions $(\mathrm{dvrl}(x_i, y_i, (y_i - f_X(x_i))) \to \mathbb{R})$, when only predictions are available $(\mathrm{dvrl}(x_i, f_X(x_i)) \to \mathbb{R})$, and when both are not available $(\mathrm{dvrl}(x_i) \to \mathbb{R})$. We use $X$ to train the predictor model and the mesh grid $X^M$ as target set.

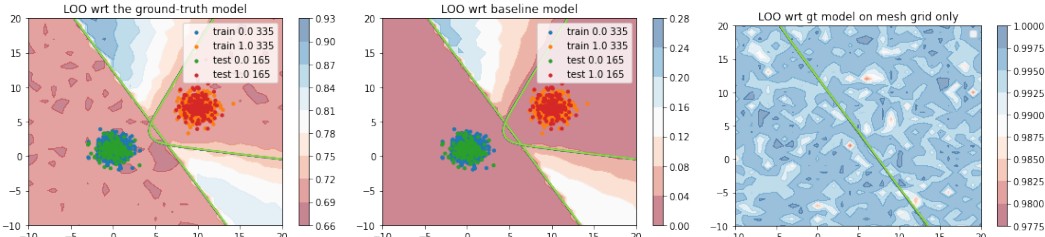

Figure 1: Data valuation using leave-one-out valuation. The left image shows a setting where the decision boundary difference between ground-truth and learned boundary is used. The center image where the difference is computed w.r.t. a baseline model, and the right image where only the mesh grid is evaluated w.r.t. to the ground-truth decision boundary. The straight line is always the ground-truth decision boundary **g** and the curve is the learned decision boundary. In the first two cases, important points are clearly in the area between ground-truth and learned decision boundary. Hence, they are the miss-classified points. In the right plot all points seem to be equally important except for some noise.

## 3 RESULTS

We present our results as color plots of data values on the mesh grid. To save some run-time, the grid for the plots is smaller than the one for comparing decision boundaries. In the following $X^{M_P}$ will refer to the smaller mesh grid for plotting and $X^M$ to the larger one for comparing the difference between decision boundaries. The plotting interval for $X^{M_P}$ is $[-10, 25]$ with a step size of one. Specifically, $X^{M_P} = \{-10, -9, ..., 25\} \times \{-10, -9, ..., 25\}$. In the appendix we further provide plots for out-of-distributions methods in Section A.2, memorization in Figure 15 and valuation of the train data in Figure 10.

### 3.1 LEAVE-ONE-OUT

First, we estimate data values on $X^{M_P}$ using leave-one-out valuation and the schemes described earlier. The results are displayed in Figure 1. Blue areas have a high data value. The left most plot shows $\mathrm{dv}(f_{X \cup x_i^{M_P}}, \mathbf{g}, X^M)$ with the ground-truth decision boundary **g** as reference and when points on the mesh grid are added to the training data. The center plot shows $\mathrm{dv}(f_{X \cup x_i^{M_P}}, f_X, X^M)$ (same as before but w.r.t. a baseline model) and the right most plot $\mathrm{dv}(f_{X^{M_P} \setminus x_i^{M_P}}, \mathbf{g}, X^M)$. The straight line is the ground-truth decision boundary **g** and the curve is the learned decision boundary. From the left and center plot it emerges that the important points w.r.t. $X$ lie in the area between ground-truth and learned decision boundary. Hence, they are the wrongly classified points. The closer a miss-classified point is to the ground-truth decision boundary the more important it is. It is noteworthy that both these plots show the same regions to be important. Hence, knowledge about the ground-truth decision boundary does not bring any benefit here. Surprisingly, both plots do not show the small miss-classification region between the clusters to be important (intersection of the decision boundaries on the interval $[5, 7]$ on the $x$-axis and $[3, 5]$ on the $y$-axis). This could, however, simply be related to the small size of the region not resulting in a large change of the decision boundary. It is also worth mentioning that some correctly classified points close to the learned decision boundary actually push this boundary away. However, this only holds for points very close to it. Finally, in the right image all regions have a similar data value. This is unsurprising because the presence of neighboring points in the mesh grid compensates for a missing point.

### 3.2 DATA SHAPLEY

Next, we use the same setup with data Shapley. Results are plotted in Figure 2. Noteworthy, the mesh grid in these experiments is smaller (step size 3) due to the large run time of Shapley. Hence, the resolution is not as fine grained. Despite that, the results are similar to those in Figure 1. Important points are those in the miss-classified region between ground-truth and learned decision boundary. Again, the left image shows a setting where $f_{X \cup x_i^{M_P}}$ is evaluated against **g** and the center plot shows

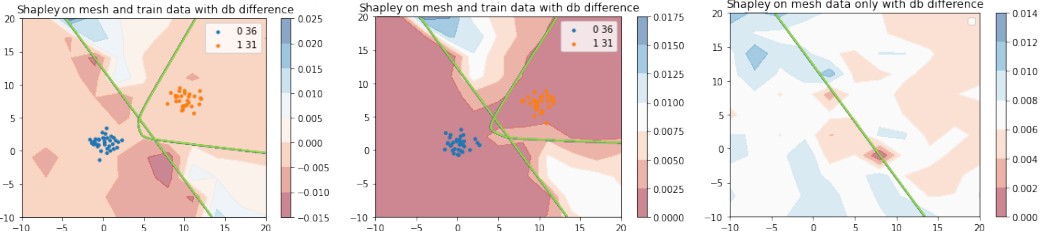

Figure 2: Data valuation with data Shapley. On the left, we again use $X$ as baseline and sequentially add points $x_i^{M_P} \in X^{M_P}$ to estimate their data value w.r.t the ground-truth model **g**. In the center image, we use the same setup with $f_X$ as reference, and, in the right image estimation is done on the mesh grid only. In the left two plots, the regions roughly match those from Figure 1 where important points are miss-classified points. In the right most plot there seems to be a small bias towards points left of the decision boundary.

$f_{X \cup x_i^{M_P}}$ against $f_X$. In the right most image the data value is measured on the mesh grid only and using the ground-truth decision boundary as reference. From this experiment it seems that Shapely has no benefit over leave-one-out valuation at least on this data set.

### 3.3 SAMPLE FORGETTING AND MEMORIZATION

The results for forgetting, last learned and memorization are plotted in Figure 3. In the left most plot data values are estimated using standard catastrophic forgetting as proposed by Toneva et al. (2018) and Algorithm 1 from the appendix. The center plot shows the setting we refer to as last learned, where we use the epoch a sample was correctly classified in for the first time as data value. In the right plot, data values are estimated with memorization events. In the left and center plot the miss-classified region between the two clusters is frequently forgotten (interval $[5, 7]$ on the $x$-axis and $[3, 5]$ on the $y$-axis) and as such important for learning. The region in the top is found to be more important when using catastrophic forgetting, while the region in the bottom is not recognized by any of the two. This is somewhat in line with the previous results since the high value points lie in one of the miss-classified regions. Sample forgetting just does not find all of these regions accurately. In the right plot, memorization finds all miss-classified regions accurately and assigns the same data value to each point in these regions. In Appendix A.7 we further provide examples when data values are evaluated on the mesh grid only and with other types of forgetting events. In general, results are similar the the ones presented here. The only difference is Figure 15 where values are computed on the mesh grid solely. In this case, points close to the ground-truth decision boundary are considered as important but we assume that this is an effect related to batch-wise training. In Figures 16, 17 and 18 we include plots that show the trajectory of the decision boundary over several epochs on some data sets. There it emerges that the forgotten points are mostly the points close to the learned decision boundary.

### 3.4 DATA VALUATION USING REINFORCEMENT LEARNING

Using DVRL we perform three different experiments with different inputs: when labels and residuals are available, when predictions are available and when only $X$ is available. The results are reported in Figure 4. DVRL uses TensorFlow (Abadi et al., 2015), different optimization parameters should therefore explain the small difference in the shape of the decision boundary compared to the previous two experiments. In all three cases, the estimated data values do not look similar to the previous results. DVRL learns to separate the blue cluster somewhere, but it does not appear to be related to the decision boundaries. It seems that DVRL distills the data set (e.g., learns to thin out $X$ by removing data points because they are over represented). In the appendix in Figure 20 we include plots after running three iterations of DVRL and find that it produces different results. We believe that this supports the hypothesis that DVRL thins out the data set by selecting some high value points at random. However, which points are selected is not deterministic.

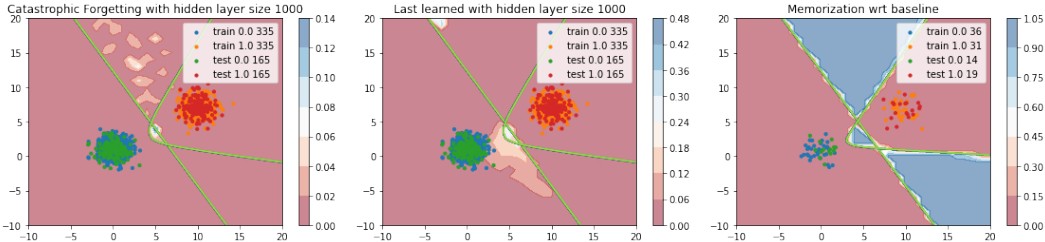

Figure 3: Data valuation using sample forgetting, last learned and memorization. The left plot shows results for standard catastrophic forgetting. The center plot shows results for latest learned data values and the right plot shows results with memorization. In the two left plots, especially points in the miss-classified region between the two clusters, interval $[5, 7]$ on the $x$-axis and $[3, 5]$ on the $y$-axis, are recognized as important. Some points in the upper miss-classified region are also considered as important but not points in the bottom region. In the right plot memorization almost exactly finds all miss-classified regions.

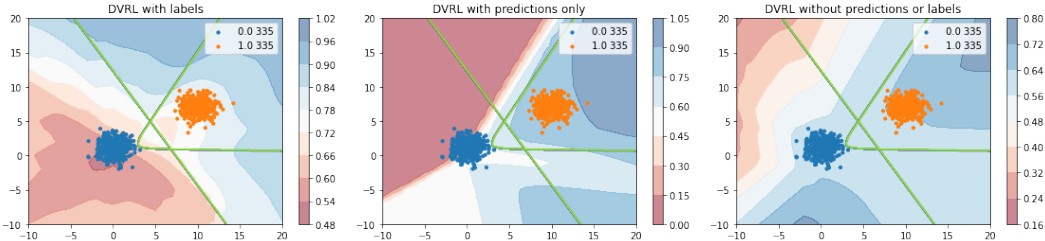

Figure 4: Data valuation using reinforcement learning. Left when labels and residuals are available $(\mathrm{dvrl}(x_i, y_i, (y_i - f_X(x_i))) \to \mathbb{R})$, center with only predictions $(\mathrm{dvrl}(x_i, f_X(x_i)) \to \mathbb{R})$, and right when both are not available $(\mathrm{dvrl}(x_i) \to \mathbb{R})$. In all cases, the learned boundary does not match the previous results. DVRL seems to distill the data only.

To summarize our results so far; leave-one-out, Shapley, sample forgetting and memorization roughly seem to agree that miss-classified points are the important points. Qualitatively, memorization seems to estimate these regions most reliably. Only DVRL deviates significantly. We use the remainder of this paper to discuss these results in more detail and to relate them to the recent literature.

## 4 DISCUSSION

Our goal was to get an intuition for the importance of data points on the performance of machine learning models. We used a synthetic 2D data set and visualized the computed data values of four different valuation methods. Data Shapley and leave-one-out valuation produce similar results. Both find the regions between ground-truth and learned decision boundary to be important. Hence, miss-classified points are the important points. Sample forgetting is in line with these results but does not seem to find all of the miss-classified regions. Memorization qualitatively produces the best results and finds all of these regions as important. In contrast to that, DVRL does not seem to produce meaningful results. Note, that this does not mean that DVRL does not work in general. In several previous experiments we successfully used it to remove unnecessary points from the training data, but it does not seem to generalize to unseen data on the mesh grid. However, DVRL has not seen the points $x_i^M$ on the mesh grid during training. In Figure 21 (see Appendix) we therefore repeat one experiment where we have added a miss-classified point to the training data of DVRL and find that a single point might already improve the generalization.

We further provide results of data valuation on the mesh grid only. For leave-one-out and Shapley (right most plot in the Figures 1 and 2) we find no special regions of interest. In sample forgetting and memorization in Figure 15 (see Appendix) the points close to the ground-truth decision boundary

seem important. We assume that this is related toggling during training. Finally, we have observed that the data values slightly differ with different data set sizes. Hence, there is an additional effect of the data set size that we do not discuss in more details (for instance, Figure 9). However, the finding that miss-classified points are the important points is constant across all experiments.

Although this finding does not seem spectacular, we were surprised by it. Our intuition has guided us to think that points close to the decision boundary would be more important and we were expecting to see out-of-distribution points to have an impact on the decision boundary as well. Similar to the support vectors in the support vector machine (SVM), we would have expected that these points would push the decision boundary further away than it is actually the case. In the appendix we repeated some of the experiments on more complex sets of data with the same results. In the remainder of this section we discuss the results and further incorporate them into the recent literature.

**Models converge to Data Set**    We were rather surprised by the finding that points close to the decision boundary only have a high influence if they are miss-classified. Otherwise, they do not seem to distract the model much, which we believe is counter intuitive. Toneva et al. (2018) have a similar intuition and compare forgettable points to support vectors. They draw a connection to Soudry et al. (2018) and see evidence that stochastic gradient descent converges to solutions maximally separating the data set. Our own results seem to contradict this theory. Correctly classified points only have an influence on the decision boundary if they are very close to it (at least on this data set). In the appendix we perform several experiments where we visualize this behavior. In particular, in Figure 9 we plot the effect of a new data point on the decision boundary for several different data set sizes. While the data point seems to introduce more variance to the model, it does not move the decision boundary as far away as we would expect for maximizing the margin.

From our experiments it seems that the majority of the data set *favours* a certain decision boundary and that a single point does not seem to have a big impact on this unless it is miss-classified. Both Toneva et al. (2018) and Paul & Dziugaite (2021) hypothesize that the importance of points is a property of the data rather than the model. We believe that our experiments support this theory. First, all models converge to a similar decision boundary in our experiments; a curve open to the right side of the plots. This does not only hold for the MLP with a single hidden layer of different size, but also to deeper models. Furthermore, in the experiments with sample forgetting in Section 3.3 and in the Appendix, the miss-classified points are considered as important. Although only a single point $x_i^M$ is added to $X$ at a time. Points in the miss-classified region seem to be harder to learn. Interestingly, they are also among the latest learned samples. That is, they are correctly classified in a later epoch during training, regardless of the used model. However, smaller models seem to have more difficulties learning these points.

**Decision Boundary Change is Enough**    Our results further show that there is no benefit of using the ground-truth decision boundary $\mathbf{g}$ over the baseline decision boundary $f_X$ to estimate high value points. Hence, for identifying important data points it is not necessary to know the ground-truth decision boundary as long as the available data is free of label noise. In other words, the data value depends on any (large) change in the learned decision boundary no matter if measured w.r.t. an arbitrary baseline or the ground-truth.

**Are no Points important by Nature?**    As mentioned earlier, we were expecting to find points close to the decision boundary to be important but our experiments contradicted this hypothesis. On the first sight, this might sound strange that a point far away from the ground-truth decision boundary may support learning this boundary. It becomes more plausible when viewed from the perspective of an SVM. According to Jacot et al. (2018), our MLP's decision boundary converges towards a corresponding linear SVM for infinitely wide perceptron layers. While this connection is not justified analytically for our finite width MLP experiments, it might still help to gain a qualitative understanding of the observed behavior. Consider a linear ground-truth decision boundary $\mathbf{g}$ and a linear SVM $f$. Given a data point $x_i$ with distance $d_i$ to $\mathbf{g}$. What is the most valuable point $x_j$ w.r.t. to $x_i$ s.t. the resulting model $f$ is close to $\mathbf{g}$? It is the point $x_j$ on the other side of the decision boundary with the same distance $d_j$ to $\mathbf{g}$ and $|x_i - x_j|_2 = d_i + d_j$. Hence, there is no single set of important points producing a perfect decision boundary, but for any point $x_i$ there is a counter point $x_j$ instead. Something similar seems to be the case for our MLP. In order to truly grasp this behavior, we propose to report data values pairwise.

**Benefits of Data Shapley and Pair-Wise Data Values**  In our experiments data Shapley seems to bring no benefit when comparing the high value data areas between the different methods. This is different from, for instance, Yoon et al. (2019) and Ghorbani & Zou (2019) where Shapley performs better than leave-one-out in most cases. However, this may be explained with the simplicity of our synthetic data set and with the sparse mesh grid. Most points on the mesh grid are singleton and therefore are not represented in any other cluster. But what should be the value of a point if it falls into a cluster or sub-population that is small? We believe that a point from a small cluster should have a higher data value. Otherwise, the smaller cluster could, for instance, be more sensitive to miss-labeled data. Leave-one-out would only find singleton points to be important but not points from a small cluster. Shapley would find points from a smaller cluster to be more important but would not reveal the cluster structure in the data (e.g., Figure 11 in the Appendix). It emerges that a large amount of points can be removed from a cluster or sub population without affecting the model performance, while it does not matter which points exactly are removed. This is in line with the idea of core-sets. However, there is not a single core-set, but many. This is different from the statement in the introduction where we hypothesized that points from a core-set should all have a high data value. A core-set should include high-value points (rare points) but also a sufficient number of points representing each cluster or sub population. Hence, for truly capturing the value of a data point we believe this point has to be considered relative to all other points. We, therefore, propose to report data values pairwise. This would give rise to some sort of adjacency matrix, reveal clusters in the data and provide a bridge to core-sets. Let $\mathbb{P}(X)$ denote the power set of $X$, a pairwise data-value could then be derived with a small extension to Shapley and would be given as $\mathrm{dvp}(x_i, x_j) = \sum_{T \in \mathbb{P}(X)} \sum_{k=0}^{|T|} |f_T(x_k) - f_{T \setminus \{x_i, x_j\}}(x_k)|$.

**Data-Centric AI and Long-Tail Theory**  Recently, the topic of data-centric AI has emerged. It extends the understanding of machine learning from considering only the model to the data used to train it. This includes quantifying and understanding the influence of data on the model and improving the data set. Jiang et al. (2020) use memorization to characterize a data set and thereby detect possible shortcomings. Feldman (2020b) uses memorization of rare samples to understand model generalization. Our own results indicate that all these methods have more in common than one would expect. Our setup may be interpreted as a Gaussian center distribution and the mesh grid as a long tail. The miss-classified singleton samples on the mesh grid are both the frequently forgotten ones as well as the ones considered as important by Shapley, leave-one-out and memorization.

For improving the underlying data set we can think of two applications. A common approach in active learning is to select samples predicted with low confidence for labeling. Our own results would suggest that this scheme should be extended to miss-labeled points. That is, the points selected for labeling should both be out-of-distribution and produce a miss-classification. If data for labeling is not available, a common approach is to use generative models such as Generative Adversarial Nets (GAN) (Goodfellow et al., 2014). In this case, the GAN could try to produce instances that would be miss-labeled by the model.

**Summary**  We performed several experiments on synthetic data and found that most methods agree on miss-classified points to be important. We focused on simple, linear and noise-free data and used only a one-hidden-layer MLP for evaluation. Our experiments are no benchmark of the evaluated methods nor do we derive a qualitative statement about them. Our goal was only to get a better understanding of data values. The next steps would include repeating our experiments on more complex data sets and describing these rules formally should they hold true. We further plan to develop a web-tool similar to the tensorflow-playground[1] for educational purpose and to perform more experiments.

**Reproducibility and Ethics Statement**  All experiments were repeated several times and we will provide the final code on GitHub. Furthermore, we tried to keep the code simple and small so that it is easy comprehensible. As our experiments were conducted on small and synthetic data and running on local machines only, the environmental impact is low. The same holds for ethical concerns, we do not see any special threats in combination with our experiments. However, we hope that our insights help to design better and more robust data sets in future.

---

[1] https://playground.tensorflow.org/

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

# A    APPENDIX

The following appendix contains additional material to support the proposed experiments.

## A.1    SYNTHETIC DATA SET

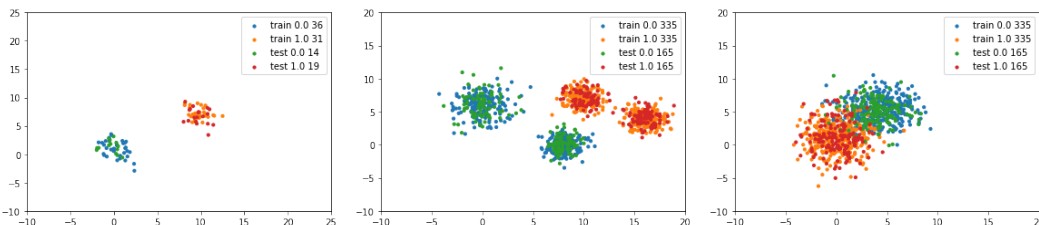

Figure 5: Snapshot of the synthetic data set for two clusters with means $(0, 1)$ and $(10, 7)$ in the left plot, with four clusters in the center, and with overlapping (noisy) clusters on the right.

The data set $X$ is two dimensional and binary. Each of the two label classes is sampled from a mixture of several Gaussian's. That is, there are $k_p$ clusters $(\mu_i^p, \Sigma_i^p)_{i=1}^{k_p}$ that represent the data with positive label and $k_n$ clusters $(\mu_i^n, \Sigma_i^n)_{i=1}^{k_n}$ that represent data with negative label. The data generator can also assign ground truth labels using one-nearest-neighbour classification. An unlabeled data point $x_i$ is assigned the label of the closest cluster mean, i.e., if the mean $\mu_i^p$ closest to $x_i$ belongs to the positive class, $x_i$ will be labeled positive. We use $\mathbf{g}(x)$ to refer to the ground truth decision boundary defined by this data generator.

### A.1.1    PARAMETERS

The mesh grid $X^M$ that we use to measure the decision boundary difference covers the range $[-10, 20]$ on both $x$ and $y$ axes and contains 500 points on each axis. Hence, $X^M \in \mathbb{R}^{500^2 \times 2}$. For plotting we use a smaller mesh grid $X^{M_P}$ in the same range. The step size for plotting is one for all experiments except Shapley. For Shapley we use a step size of three because of the large run time. That is, the values on $x$ and $y$ axes are either $\{-10, -9, ...20\}$ or $\{-10, -7, ..., 20\}$. Exemplary, $X^{M_P} = \{(-10, -10), (-10, -9), (-10, -8)..., (20, 20)\}$.

## A.2 OUT-OF-DISTRIBUTION RECOGNITION

We started with a use case where we aimed to estimate data values of unlabeled points. Our first intuition was therefore to exploit out-of-distribution methods. The results of data valuation using three common methods are presented in Figure 6. The blue areas represent regions with a high data value and red regions accordingly with a low value. The two green lines are the ground-truth decision boundary (straight line) and the decision boundary representing the MLP (curved line). Both Monte Carlo (MC) dropout and the ensemble only recognize points close to the decision boundary as OOD while the Gaussian process detects all points outside the train set distribution as OOD. We briefly revisit ensemble methods and Monte-Carlo dropout below.



Figure 6: Data valuation using three common out-of-distribution (OOD) methods. Blue areas have a high data value (are OOD) while red areas are less important. The Gaussian process (left plot) recognizes all points outside of the train distribution as OOD while Monte Carlo dropout (center plot) and ensemble methods (right plot) recognize only points close to the decision boundary. Monte Carlo Dropout is better in finding points in the upper part of the decision boundary.

**Ensemble Methods** Ensemble methods (Lakshminarayanan et al., 2017) work by training several different models on the same task. The uncertainty is estimated using the variance in the prediction of all these methods. Intuitively, this variance is high for points unseen during training. In the section A.3 we performed some experiments on how to select an appropriate model size. In general, smaller models are better for this purpose.

**Monte-Carlo Dropout** A common method for out-of-distribution recognition in neural networks is Monte-Carlo Dropout (Gal & Ghahramani, 2016). The idea is rather simple, dropout is using during inference. That is, inference is run several times on the same model with dropout. The variance in the prediction is then used as uncertainty. This method has the advantage that it does not require training several models.

### A.3 MODEL STABILITY EXPERIMENTS

This section shows some of the initial experiments we performed in order to find stable architectures. Figure 7 shows the model stability of different architectures of a single MLP with different hidden layer sizes. On the other hand, Figure 8 shows the model stability when a new point appears in the center of the cluster. Finally, Figure 9 shows the model stability with different data set sizes when a new point is added.

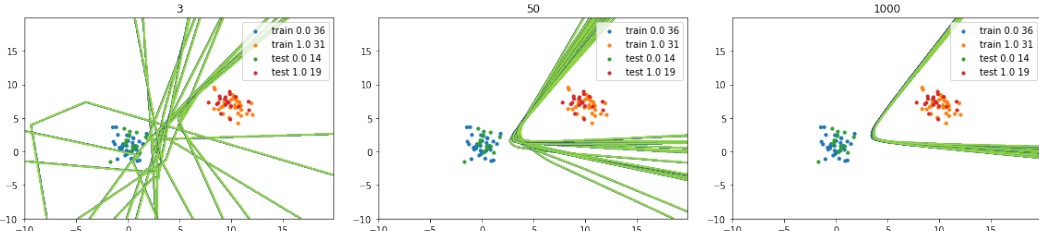

Figure 7: Model stability for different architectures of a single layer MLP with hidden layer size 3, 50 and 1000 for 15 training runs. Each line is the decision boundary after a full training. Larger models produce more stable decision boundaries.

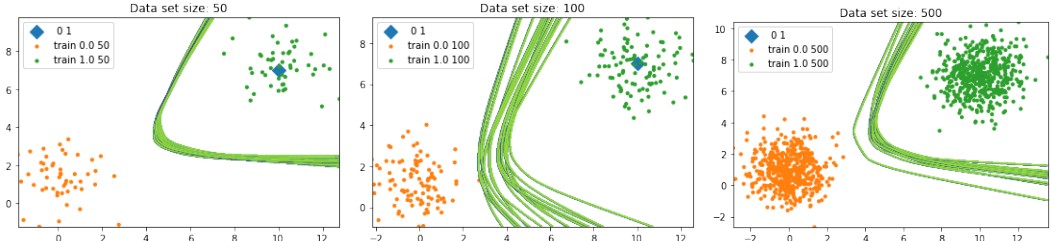

Figure 8: Stability of models when a new miss-labeled point (blue square) appears in the center of the cluster. Larger data sets seem to be distracted more. The hidden layer size of the MLP is 1000 in these experiments. Hence, without the miss-labeled point we would expect the same stability as above. While the miss-labeled point does not have an impact on the predictions in the train data, it induces noise to the decision boundary, which might be problematic.

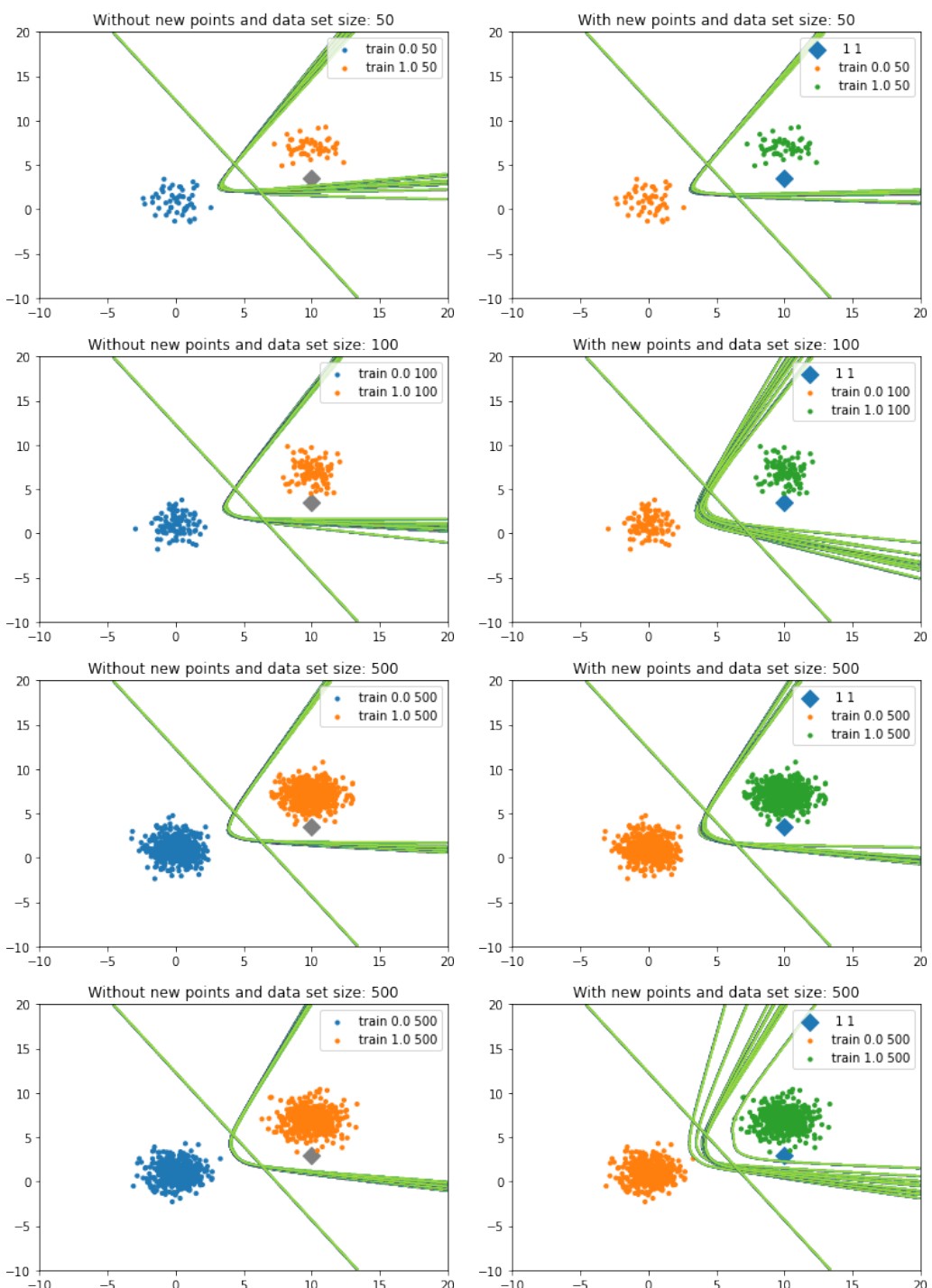

Figure 9: Model stability for different sizes of the training set when a new correctly labeled point (blue square) is added to the data set. The left column shows the data set without the new point (therefore, it is grey) and the right one with new point. The first three rows show data set sizes of 50, 100 and 500. The forth row shows a size of 500 again. As before, for larger data sets a new point seems to cause more variance in the decision boundary. In the top three rows this variance is moderate. In the last row it is, however, larger although it is the same setup as in row three except that the data set was re-sampled. Apparently, some initializations are more prone to noise. We did not observe such noise in the experiments in the paper.

## A.4 SAMPLE FORGETTING

Apart from the standard definition of catastrophic forgetting given in the Algorithm below, we compare several deviations which we call *last learned*, *simple forgetting* and *frequent forgetting*. For last learned we count the first epoch an instance was correctly classified in as data value. Frequent forgetting counts forgetting events whenever a sample was classified correctly beforehand, and simple forgetting does not require a point to be classified correctly before.

**Data:** $D = \{(x_i, y_i)\}_{i=0}^{n}$, $D^M = \{(x_i^M, y_i^M)\}_{i=0}^{m^2}$, num_epochs, Model $f(x, \theta)$
**Result:** $dv \in \mathbb{R}^{M^2}$

$dv \leftarrow \{0\}^{M^2}$;

**for** $(x_i^M, y_i^M) \in D^M$ **do**
    $X_{new} = X \cup x_i^M$;
    $y_{new} = y \cup y_i^M$;

    prev_acc $= 0$;
    forgetting_stats $= 0$;

    **for** $k$ *in* num_epochs **do**
        fit $f$ on $\{X_{new}, y_{new}\}$ for one epoch;
        $\hat{y}_i^M = f(x_i^M)$;
        $acc = 1$ **if** $\hat{y}_i^M = y_i^M$ **else** $0$;

        **if** $acc \leq$ prev_acc **then**
            $dv[i] \leftarrow dv[i] + 1$
        **end**

        prev_acc $= acc$;
    **end**

    $dv[i] \leftarrow \frac{dv[i]}{num-epochs}$
**end**

**Algorithm 1:** Data Valuation with Catastrophic Forgetting.

## A.5  DATA VALUES ON TRAINING DATA

Although it is not the main goal of the paper, we provide some results of train data valuation in this section. Figure 10 shows the valuation of the training data using leave-one-out, Shapley, last learned and DVRL (with labels). A similar experiment with a more complex data set is instead depicted in Figure 11. In this case, the methods were applied to a data set with four clusters.

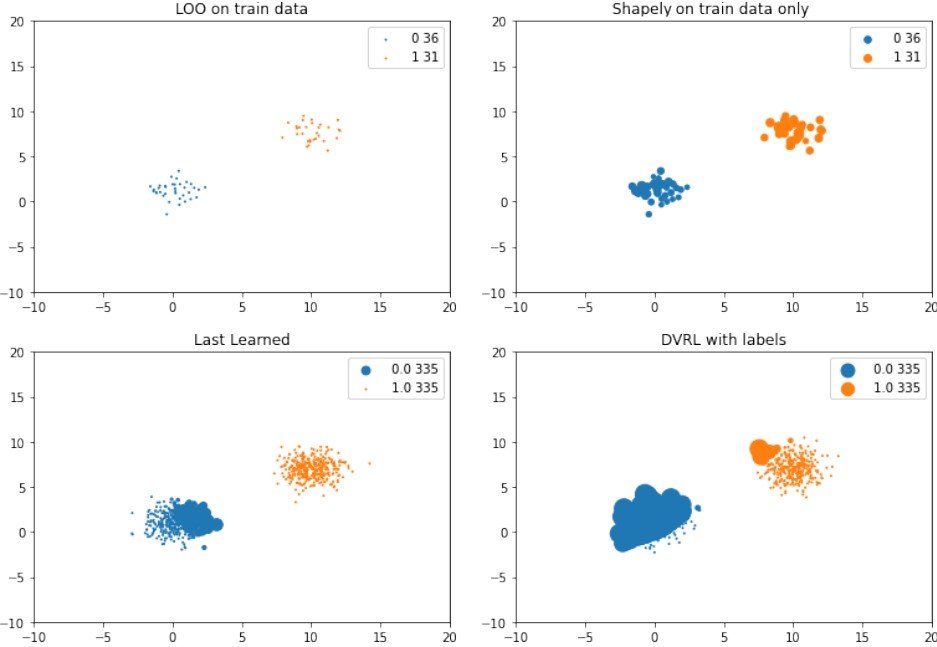

Figure 10: Valuation of the training data. The size of a point represents its data value. Leave-one-out (top left) and catastrophic forgetting (shown in Figure 16) assign the same data value to every point. Shapley (top right) shows only minor differences on the point sizes. Only last-learned samples and DVRL produce different results. If data values are used to remove points from the data set, DVRL and last learned produce the best results.

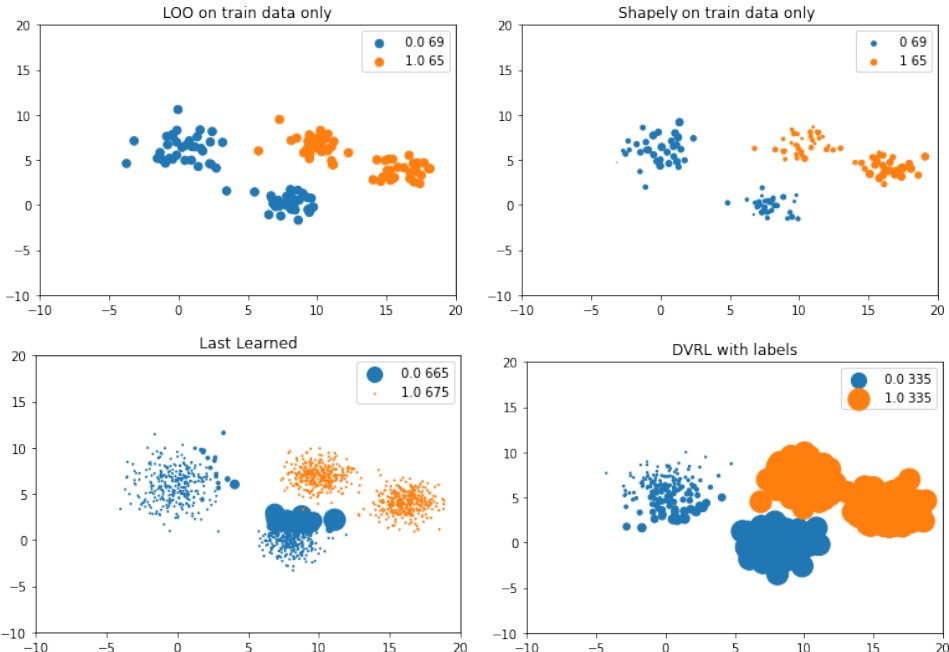

Figure 11: Data valuation of training data with leave-one-out, Shapley, latest-learned and DVRL on data from four clusters. Again, leave-one-out (top left) assigns almost the same data value to each point. Shapley (top right) shows more variance this time and values one of the two clusters for each label slightly higher than the other. The results for catastrophic forgetting are visible in Figure 17. As mentioned in Figure 17, it seems to find points close to the decision boundary to be most important. Last learned and DVRL again show the most variance in the data values and consider one cluster as much less important than the others. Again they produces the best results if the goal is to reduce the size of the data set.

### A.6   OTHER DATA SETS

To validate our findings in the main paper, we have repeated the experiments on three other data sets. The results are reported in this section. Figure 12 shows the results of data valuation on noisy data. Figure 13 shows the results of evaluating data from four clusters. Finally, Figure 14 shows the results of leave-one-out, sample forgetting and DVRL on data centered in the origin.

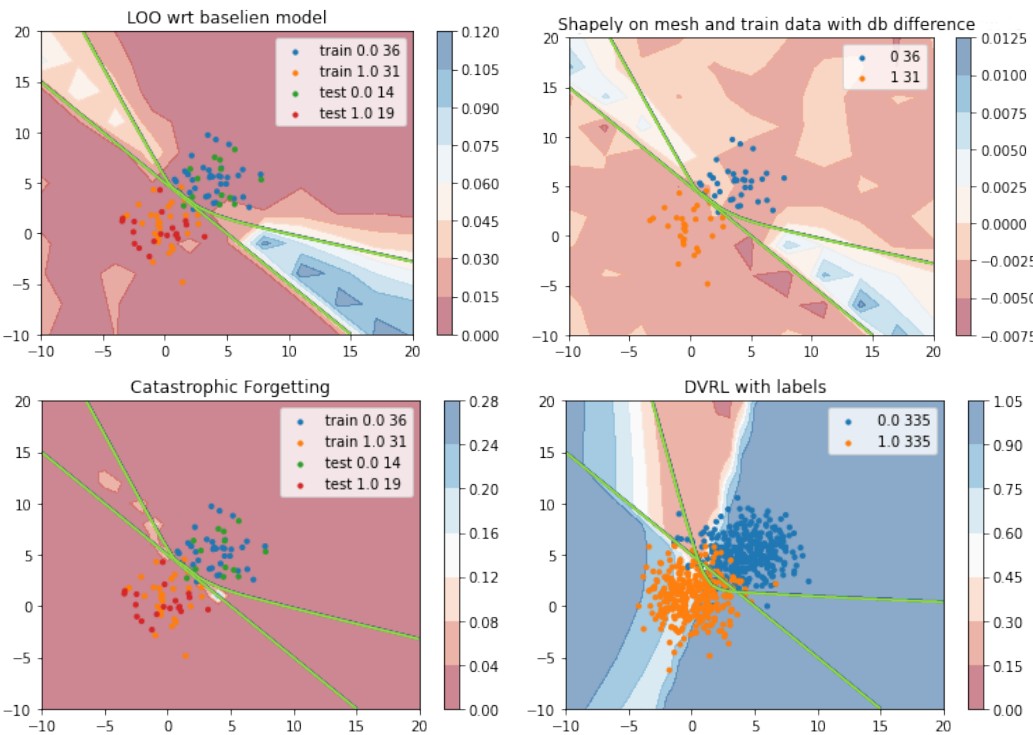

Figure 12: Results for data valuation on noisy data (overlapping clusters). Results are similar to those in the main paper. Leave-one-out and shapely (top row) largely agree on the miss-classified region to be important. Sample forgetting (bottom left) finds points between the clusters as well as part of the points in the upper miss-classified region as important. DVRL again produces different results.

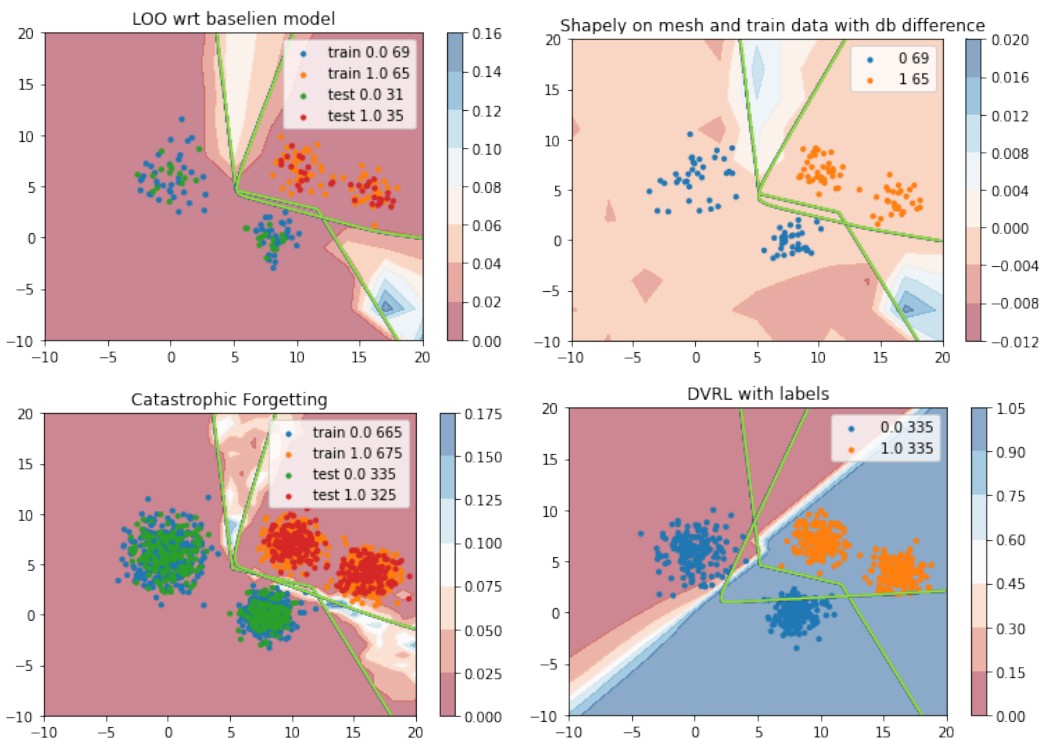

Figure 13: Results for data valuation on data from four means. Again, results are similar to the ones before. Leave-one-out and Shapley agree on the miss-classified regions as important but do not find the very small miss-classified region between clusters as important. Sample forgetting finds the region between clusters and, additionally, both other miss-classified regions as important. DVRL again falls out of the box.

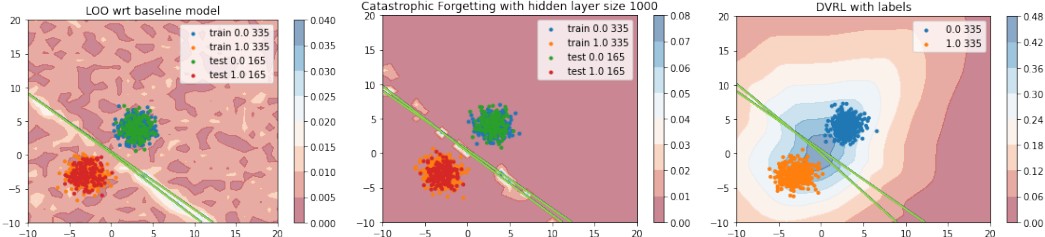

Figure 14: Leave-one-out, sample forgetting and DVRL on zero centered data. In the main paper we use data not centered in the origin because we find that our MLP thereby learns a non-linear decision boundary in the interval of interest. Here, we show results on data centered in the origin. This plot may also serve as reminder that data normalization is important. Apart from this, points in the miss-classified region and close to the decision boundaries seem more important with leave-one-out and sample forgetting. DVRL finds the region around the center including small parts of each cluster to be important. Results with DVRL seem more intuitive in this case.

## A.7 FURTHER EXPERIMENTS

This section contains some other experiments that we believe might be considered as important and do not fall in any of the categories above. Figure 15 shows data valuation with catastrophic forgetting and memorization events on a mesh grid. Figures 16, 17 and 18 show the trajectory of the decision boundary for catastrophic forgetting and importance of points as size using a data set with two and four means and noisy data, respectively. Figure 19 shows data valuation with last-learned samples, frequently-forgotten samples and simple forgetting. Finally, Figures 19 and 21 are about DVRL. The first Figure shows the different results of DVRL after three iterations. The second Figure shows the performance of DVRL with a new rare point.

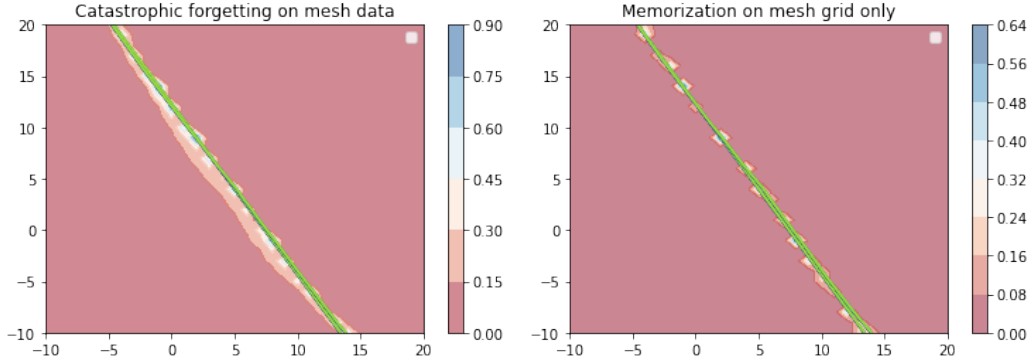

Figure 15: Data valuation with catastrophic forgetting and memorization events on mesh grid only. In this case only points next to the decision boundary are considered as important. We assume that this is most likely due to random toggling of the decision boundary as a result of batch-wise training. This is also in line with the following results in Figures 16, 17 and 18 where forgotten points are close to the learned decision boundary. In this Figure, the mesh grid already provides an almost perfect decision boundary.

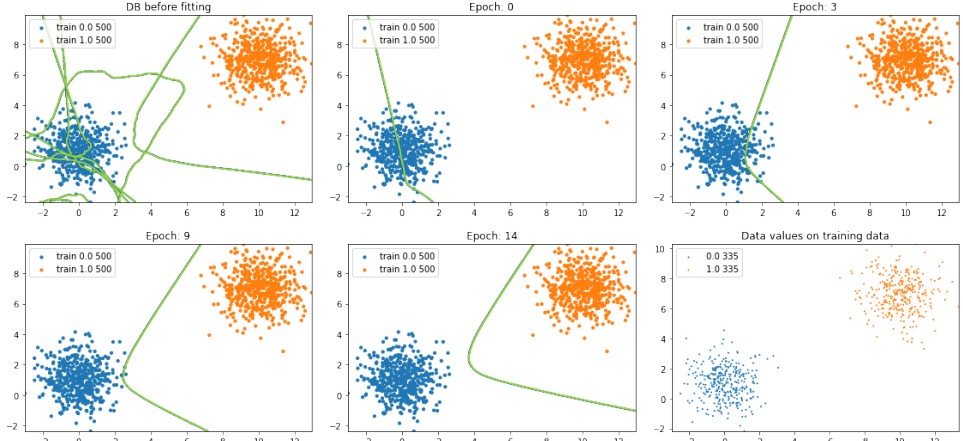

Figure 16: Trajectory of decision boundary for catastrophic forgetting and importance of points as size. The first plot shows 10 different random decision boundaries on initialization of the MLP. The next four plots show the decision boundary after the first, third, ninth and fourteenth epoch. The last image shows the final data values as size of training points. All points have the same data value here.

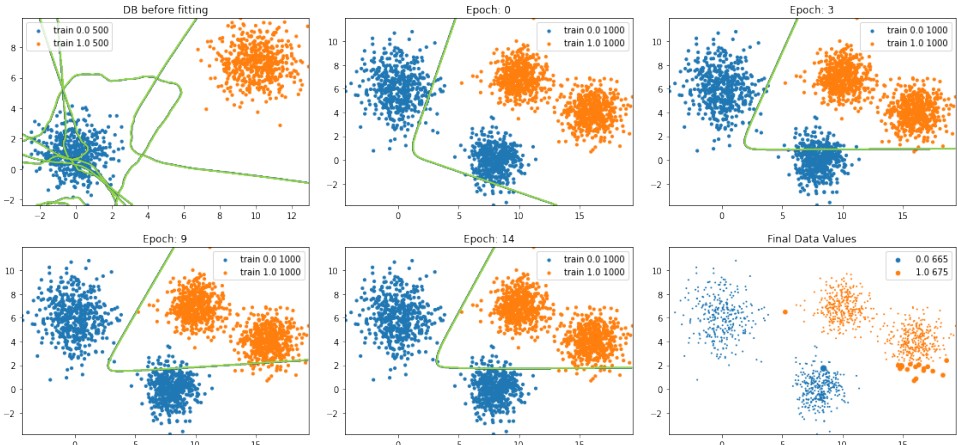

Figure 17: Trajectory of decision boundary for catastrophic forgetting on data from four means. The first plot exemplary shows 10 different decision boundaries on another data set. The next four plots show the decision boundary after the first, third, ninth and fourteenth epoch. The last image shows the final data values as size of training points. Only points close to the decision boundary have a high data value.

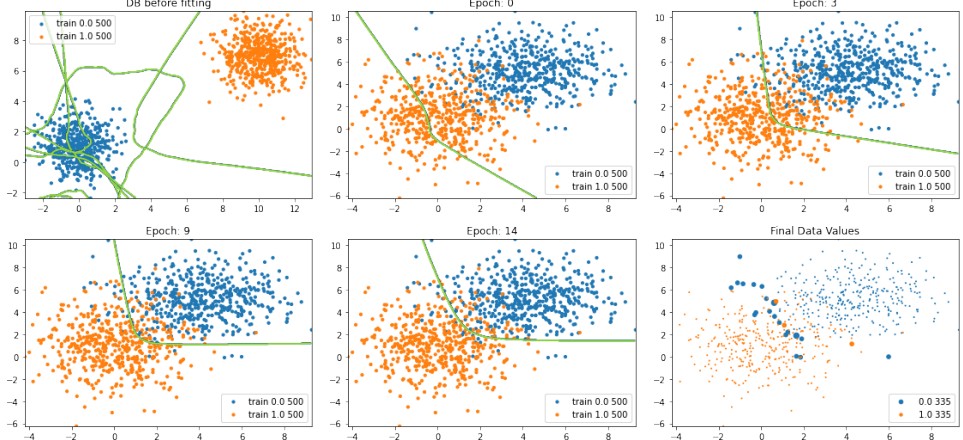

Figure 18: Trajectory of decision boundary for catastrophic forgetting on noisy data. The first plot exemplary shows 10 different decision boundaries on another data set. The next four plots show the decision boundary after the first, third, ninth and fourteenth epoch. The last image shows the final data values as size of training points. Again, points close to the decision boundary have a high data value.

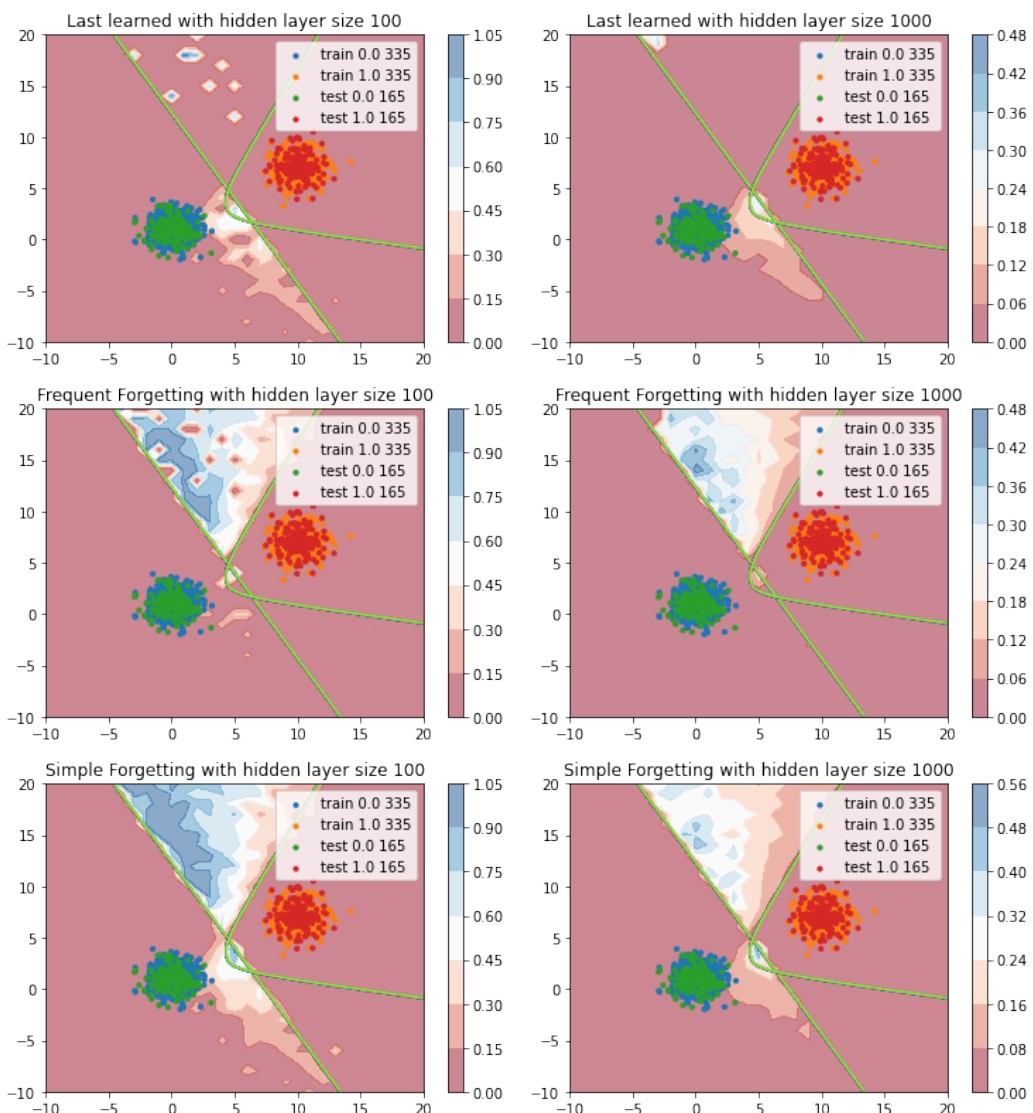

Figure 19: Data valuation with last-learned samples (top), frequently-forgotten samples (middle) and simple forgetting (bottom). For finding last-learned samples we use the epoch a point was correctly classified in for the first time as data value. Frequently forgotten samples are similar to sample forgetting in the main paper, but we only require them to be classified correctly at least once (and not in the previous epoch). Simple forgetting counts how often a point was forgotten but does not require that is was correctly classified before. Hence, never learned points count here as well. Again, the miss-classified region in the top seems to be the hardest to learn.

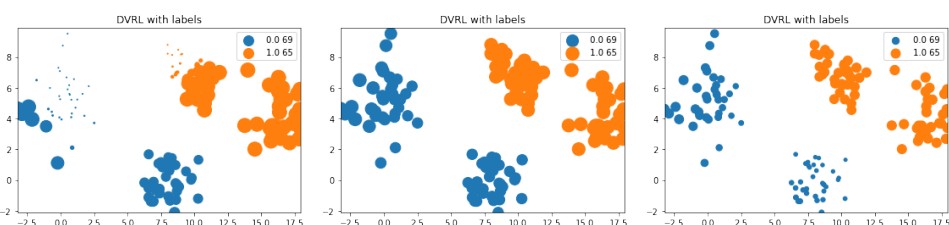

Figure 20: Three iterations of DVRL on the same training set produce different results.

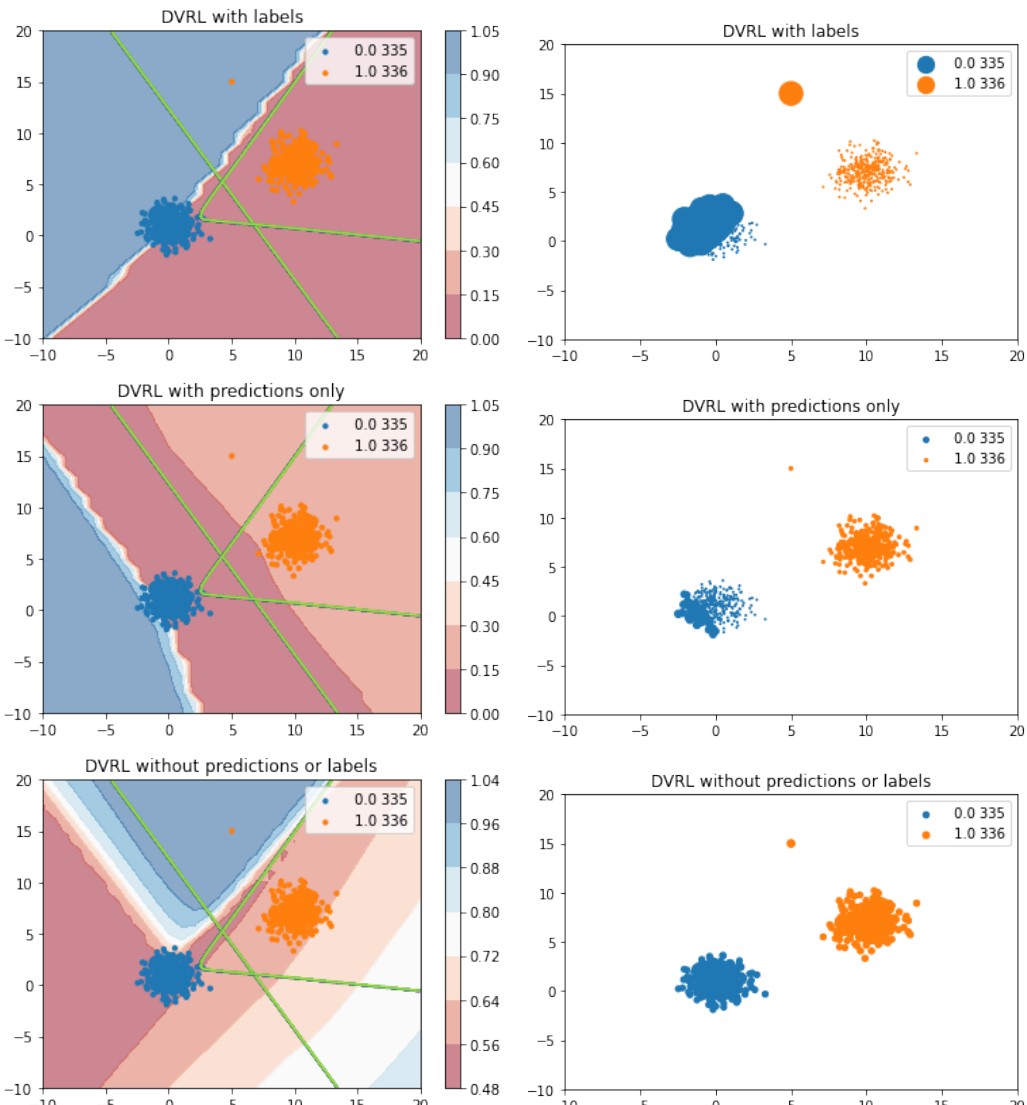

Figure 21: DVRL with new rare point. The new point is the orange dot at $(15, 5)$ and as such falls into the miss-classified region. In this case, DVRL with the full input correctly labels the point as important. DVRL with predictions only does not recognize it as important. DVRL without predictions or labels, however, recognizes it as important, and, surprisingly, finds a region close to the miss-classified region to be important.

