# OpenReview forum: "Towards Understanding Data Values: Empirical Results on Synthetic Data"
_ICLR.cc/2022/Conference — ICLR 2022 Submitted_

### Official Review · Reviewer_Dtr6 · 2021-10-30

**Correctness:** 4
**Technical Novelty And Significance:** 3
**Empirical Novelty And Significance:** 3
**Recommendation:** 6
**Confidence:** 3

**Main Review:**

The paper provides a relevant contribution on a fairly recent ML area concerning the importance of data points.
The authors show, on a simple and controlled 2D synthetic dataset, different situations and different impact of data points on the considered predictive model.
They also provide a comprehensive definition of importance for a data points, separated into two aspects that they show to be quite similar for noise-free datasets.
Throughout a suite of experiments by MLP in different setups they found that most methods agree on miss-classified points to be important.
A number of interesting figures are included showing the different impact of the added data points to the boundary shapes: however, a few tables reporting the actual numbers to quantitatively support the authors' claims would be recommended.
As the authors acknowledge, the limitation of tested models (MLPs) and datasets make the contribution just a proof of concept to be used as a basis for a larger study to lay down the foundation of a grounded theory of data importance.


**Summary Of The Paper:**

The authors introduce a novel framework to assess data points importance as the change of the decision boundary for a model with respect to added samples. They demonstrate their claim by training a MLP model on a synthetic 2D data set in several modalities and show the criticality of misclassified points.


**Summary Of The Review:**

Paper is well written, readable by a wide audience and tackling a quite relevant problem.
Experiments are convincing and supporting the authors' claims, although limited to a single model type and simple 2d synthetic dataset.
Overall, the authors contribute novel insights on a interesting ML problem,.

---

> ### Author Response · Authors · 2021-11-10
> **Reply to Dtr6**
>
> Thank you for the review and the effort, it is much appreciated. We indeed considered putting the experiments into a table to save some space in the main paper. Regarding the proof-of-concept state, we are already discussing adaptation to test the findings on real data.

---

### Official Review · Reviewer_hZMc · 2021-11-02

**Correctness:** 2
**Technical Novelty And Significance:** 2
**Empirical Novelty And Significance:** 3
**Recommendation:** 3
**Confidence:** 4

**Main Review:**

The paper is pretty straightforward. It modifies four different well-known techniques for evaluating data. The biggest weakness of the paper is that it is based on a single small size 2-dimensional synthetic dataset. The result and the conclusions are based on empirical visualizations. The results are very preliminary and limited to make any secure conclusions. The method is not practical at all for real data, since it is difficult to visualize.

The intuition and the methodology that the authors are using are in the right direction, but it is half-baked. The authors need to find a way to work it out in higher dimensions and in more real-world scenarios. One direction that the authors could explore is to measure the value of the points in terms of dollars (or any other coin) and not in terms of accuracy.

The appendix is not very useful. There are many more experiments mentioned there, but it makes it very hard to follow the paper. There are way too many plots. It would make more sense to create some measures over the plots and present them on a table.

**Summary Of The Paper:**

The paper proposes a new method for identifying important points in a dataset given the task of classification. The paper introduces a new valuation function. The paper takes into consideration other methods that deal with the same problem and reevaluated them using the new scoring function. The scoring function focuses on the effect points have on the decision boundary. The results show that the most valuable points are not the ones close to the boundary but the misclassified ones. The result is interesting but not surprising.

**Summary Of The Review:**

The approach of the paper is interesting, but the paper needs a lot of work to be a full conference paper. It makes more sense to present it at a workshop and get more feedback. As mentioned in the main review the authors present some preliminary evidence for their findings, but more experiments are needed in order to have a strong claim.

---

> ### Author Response · Authors · 2021-11-10
> **Reply to hZMc**
>
> Thank you for the feedback it is much appreciated. We will definitely consider to present the work in a workshop. Below you can find our response for each paragraph.
>
> *"The paper is pretty straightforward. It modifies four different well-known techniques for evaluating data. The biggest weakness of the paper is that it is based on a single small size 2-dimensional synthetic dataset. The result and the conclusions are based on empirical visualizations. The results are very preliminary and limited to make any secure conclusions. The method is not practical at all for real data, since it is difficult to visualize."*
>
> We are aware that experiments on synthetic 2D data do not provide any prove for higher dimensional or real world data. The reason why we still chose 2D data is that we were aiming for an intuition in a controlled and understandable environment. On real world data it is not straight forward to create the mesh-grid as we require it for our experiments and it is not easy to visualize. In our plots it directly emerges that miss-classified points are important. We did not really aim at a method applicable for real data but simply for the intuition and for some insights. Our hope was that the community would find these insights interesting enough before validating them on real world data.
>
> *"The intuition and the methodology that the authors are using are in the right direction, but it is half-baked. The authors need to find a way to work it out in higher dimensions and in more real-world scenarios. One direction that the authors could explore is to measure the value of the points in terms of dollars (or any other coin) and not in terms of accuracy."*
>
> Our motivation was to establish a more general understanding of the influence of data on the model quality. Your suggestion, however, is certainly a consequential next step since model quality is directly related to a financial value in an application context. We actually had a few ideas on applying the method on real data. First, it should be noted that applying the method to real data does not really bring any practical usage in our opinion but simply validates the findings on simpler data. One option we had in mind was to use a simple real data set such as MNIST, extract an embedding using a pre-trained CNN and down-size the embedding to 2D using PCA  . This way we could use MNIST directly in our existing pipeline and could visualize the generated 2D mesh-grid in image space using the ‘inverse_transform’ of PCA.
>
> *"The appendix is not very useful. There are many more experiments mentioned there, but it makes it very hard to follow the paper. There are way too many plots. It would make more sense to create some measures over the plots and present them on a table."*
>
> We agree that the appendix is a bit too long and contains too many experiments. We are also aware that many experiments from the appendix are referenced in the main paper, which disturbs the flow of reading.

---

> > ### Comment · Reviewer_hZMc · 2021-11-29
> > **Response**
> >
> > I would like to thank the authors for their response. Their suggestion of validating their method with PCA should be integrated into future versions of the paper. At this point, we don't seem to have a disagreement. The decision of acceptance will be determined on the sufficiency of the presented material.

---

### Official Review · Reviewer_roaw · 2021-11-02

**Correctness:** 3
**Technical Novelty And Significance:** 3
**Empirical Novelty And Significance:** 3
**Recommendation:** 3
**Confidence:** 5

**Main Review:**

Pros:

- The topic is of critical importance to the machine learning community.
- The paper is well-written and easy to follow.

Cons:
- The paper lacks theoretical justification for the findings.
- The experiments are only performed on MLP with a single layer and low-dimensional synthetic data. It is questionable whether the findings from the experiments contribute generalizable knowledge to deeper neural networks, higher-dimensional, real-world data, which is of more interest in practice.
- The paper presents a comprehensive overview of data valuation ideas. For Shapley value, though, the authors only discuss one specific algorithm, DataShapley, to calculate the Shapley value. This algorithm is a heuristic for calculating Shapley and there are no theoretical guarantees for the approximation error. Hence, it remains a question whether the findings presented in the paper are property of Shapley value or property of the DataShapley heuristic. In particular, there exist efficient algorithms for calculating Shapley value with provable error guarantees, e.g., permutation sampling and group testing in [1], for the scale of the experiments conducted in the paper. It'll be great to include these algorithms in the comparison.

[1] https://arxiv.org/abs/1902.10275

**Summary Of The Paper:**

The paper presents an empirical study of different data valuation techniques based on 2D synthetic data and discusses the findings.

**Summary Of The Review:**

The paper studies a very timely and important topic. Yet, the main concern is that the paper lacks formal justification and the experiments are done in a synthetic setting. It is questionable how generalizable the findings are beyond the specific synthetic data and the specific model considered in the paper.

--
[post-rebuttal] Thanks to the authors for the response. This is an interesting paper but my concerns about the lack of theoretical justification and experiments of modern DNNs as well as missing baselines still remain. Hence I will maintain my score.

---

> ### Author Response · Authors · 2021-11-10
> **Reply to Reviewer roaw**
>
> Thank you for your feedback it is much appreciated. Please find our response to the three cons below.
>
> *"The paper lacks theoretical justification for the findings."*
>
> We tried to provide some minor explanations between the lines, but we agree that it is not very much. Regarding the finding that miss-classified points are important we, for instance, hypothesize that the majority of the data set leads the model to converge to a specific decision boundar y. If a point is in-line with this decision boundary (is not miss-classified) it does not need any change of this decision boundary. Our main goal at this point was to share a simple finding that we considered as interesting and partially intuitive.
>
> *"The experiments are only performed on MLP with a single layer and low-dimensional synthetic data. It is questionable whether the findings from the experiments contribute generalizable knowledge to deeper neural networks, higher-dimensional, real-world data, which is of more interest in practice."*
>
> In the appendix and supplementary material, there are a few experiments with deeper models (two or more hidden layers) and different implementations (TensorFlor vs sklearn) but no CNNs. We are working on a web-app that should simplify such experiments and data exports in future. Regarding the applicability to real data, we had several ideas on how to approach this and how to validate the findings on more complex data sets. As mentioned above, our only goal here was to share a simple finding that we believed might be interesting. We left the validation of these results for future work and hoped that the initial results would be of enough interest to the community. Especially, since the application to real world data would require some changes to the setup.
>
> *"The paper presents a comprehensive overview of data valuation ideas. For Shapley value, though, the authors only discuss one specific algorithm, DataShapley, to calculate the Shapley value. This algorithm is a heuristic for calculating Shapley and there are no theoretical guarantees for the approximation error. Hence, it remains a question whether the findings presented in the paper are property of Shapley value or property of the DataShapley heuristic. In particular, there exist efficient algorithms for calculating Shapley value with provable error guarantees, e.g., permutation sampling and group testing in [1], for the scale of the experiments conducted in the paper. It'll be great to include these algorithms in the comparison."*
>
> Thank you for the hint. We were not aware of the other implementations. However, since we observed results in line with leave-one-out and other methods we do not believe that we would see significantly different results here.

---

### Official Review · Reviewer_vtgh · 2021-11-05

**Correctness:** 4
**Technical Novelty And Significance:** 2
**Empirical Novelty And Significance:** 2
**Recommendation:** 3
**Confidence:** 4

**Main Review:**

Strengths: This paper compares several methods by which to assess the value of data points for classification.

Weaknesses:
1. The results that are surprising or are contradictory with other papers in the area (e.g., adding data points to larger training sets leads to larger changes in the model, only misclassified points close to the boundary are important) are not explained in any detail. Just noting the disagreement is not helpful---an explanation, or at least a hypothesis, of why there is a disagreement is needed to give some useful insight.
2. Since Support Vector Machines are a well-established and high performing model that assesses the value of data points, comparison with SVM should be done. I realize that the goal here is to compare different methods for assessing the value of points for a particular classifier (multi-layer perceptron). However, I still feel that a comparison to SVM is valuable since it is so well established.

**Summary Of The Paper:**

This paper uses a synthetic two-dimensional dataset to visualize the importance of different data points on machine learning model performance. In particular, they used a multi-layer perceptron as the model, and they used four different schemes by which to measure the importance of individual data points. Not surprisingly, they show that regions of data that are misclassified tend to have relatively high importance in determining model performance. However, there are notable differences among the different schemes in their performances. This paper discusses the nature of those differences.

**Summary Of The Review:**

This paper compares methods by which to assess the value of data points for classification. However, insufficient analysis of surprising findings and a lack of comparison to a well-established method that assesses the value of data points (SVMs) weaken the contribution of this paper significantly.

---

### Decision · Program_Chairs · 2022-01-20

**Decision:**

Reject

**Comment:**

The reviews are of adequate quality. The responses by the authors are commendable, but ICLR is selective and reviewers continue to believe that more experiments and more rigorous analysis are needed.